# Hidden in Plain Sight: Alphavirus Persistence and Its Potential for Driving Chronic Pathogenesis

**DOI:** 10.3390/v18010030

**Published:** 2025-12-24

**Authors:** Maria del Mar Villanueva Guzman, Zhenlan Yao, Melody M. H. Li, Maria Gabriela Noval

**Affiliations:** 1Department of Microbiology, Icahn School of Medicine at Mount Sinai, New York, NY 10029, USA; mariadelmar.villanuevaguzman@icahn.mssm.edu; 2Graduate School of Biomedical Sciences, Icahn School of Medicine at Mount Sinai, New York, NY 10029, USA; 3Department of Microbiology, Immunology and Molecular Genetics, University of California, Los Angeles, CA 90095, USA

**Keywords:** viral persistence, alphavirus pathogenesis, host–virus interactions, cellular reservoirs, immune modulation, chronic viral infection

## Abstract

Alphaviruses have historically been viewed as acute, self-limiting pathogens. However, growing evidence shows that viral RNA and antigens can persist in vertebrate hosts long after the resolution of acute infection, a phenomenon known as viral persistence. Viral persistence reflects a dynamic interplay between viral replication—including shifts from lytic to non-lytic infection—and host defenses, which together establish cellular and tissue niches that enable evasion of immune-mediated clearance. Within vertebrate hosts, alphaviruses exhibit broad tissue tropism, infecting diverse cell types that may differentially support long-term persistence. Emerging evidence suggests that viral persistence arises through three interconnected processes: (i) selective infection of specific cellular niches, (ii) reprogramming of host cellular pathways, and (iii) modulation of immune responses. Yet, the extent to which viral or host determinants shape this balance, and how persistence contributes to chronic disease, remains unresolved. Here, we synthesize current *in vitro* and *in vivo* evidence of alphavirus persistence in vertebrate hosts and discuss potential mechanisms by which alphaviruses establish and maintain persistent infection beyond the acute phase. We further underscore critical gaps in current knowledge and outline future research avenues essential for elucidating the mechanisms underlying alphavirus pathogenesis.

## 1. Introduction

Alphaviruses are approximately 12-kilobase (kb), positive-sense, single-stranded RNA (+ssRNA) viruses in the *Togaviridae* family and are transmitted primarily by mosquito vectors [1]. Although traditionally classified as “Old-World” or “New-World” alphaviruses, this geographic terminology no longer reflects their rapidly expanding global distribution. In this review, we classify alphaviruses according to their predominant clinical manifestations as arthritogenic alphaviruses—including chikungunya virus (CHIKV), Mayaro virus (MAYV), Ross River virus (RRV), Sindbis virus (SINV), Semliki Forest virus (SFV), and o’nyong-nyong virus (ONNV)—or encephalitic alphaviruses, such as Venezuelan equine encephalitis virus (VEEV), eastern equine encephalitis virus (EEEV), and western equine encephalitis virus (WEEV) [2]. Alphaviruses represent a group of human pathogens with significant epidemic potential.

In recent decades, there has been an unprecedented rise in alphavirus infections worldwide [3,4,5], driven by the increasing migration of humans into new ecological zones, the climate-driven geographical expansion of mosquito vectors such as *Aedes aegypti* and *Ae. albopictus* [5,6,7], and the emergence of viral variants with expanded **vectorial capacity** (see Glossary) [8]. CHIKV is one of the most striking examples. Before 2004, it circulated sporadically in Africa and Asia, causing only hundreds to thousands of cases annually [9]. Since then, CHIKV has spread to over 100 countries, with an estimated 35 million cases per year [4]. As of October 2025, outbreaks with autochthonous transmission have been continuing across Europe, the Americas, Africa, and Asia, including a re-emergence in mainland China since July 2025 [10,11]. These trends, together with the first locally acquired CHIKV case in New York in 2025 [12], underscore that temperate regions are no longer exempt from transmission.

Concerning trends are being observed for other alphaviruses as well. MAYV incidence is rising in Brazil [13,14], and imported cases have been documented in Europe [15,16,17,18]. Although primarily transmitted by *Haemagogus* spp. mosquitoes in sylvatic areas, experimental studies show that MAYV can also be transmitted by *Ae. aegypti*, *Ae. albopictus*, and *Anopheles* species, raising concerns about urban spillover [16,19,20,21,22]. Additionally, encephalitic alphaviruses, primarily transmitted by *Culex* spp., continue to circulate in the Americas, causing sporadic outbreaks affecting thousands [23,24,25,26]. In 2024, 19 human EEEV cases and five deaths were reported in the United States, and the re-emergence of WEEV in Argentina and Uruguay in 2023 suggests that shifts in bird migratory patterns and mosquito distribution are expanding regions at risk [5]. Together, these trends highlight the increased vulnerability of naïve populations to emerging alphavirus threats.

Alphavirus infections cause a broad spectrum of acute symptoms, ranging from mild, self-limiting febrile illness to debilitating clinical manifestations and even death. Arthritogenic alphaviruses are symptomatic in 80–90% of cases, typically presenting with fever, rash, and joint pain that resolve within days to weeks [27]. In a subset of cases (0.6–13%), particularly in infants, older adults, and individuals with comorbidities, viruses such as CHIKV can also cause severe complications requiring hospitalization, including cardiovascular, neurological, and hepatological manifestations [27,28,29]. The mortality rate associated with CHIKV has been reported to be 0.1% in symptomatic infections [30] and can rise to 5% among hospitalized individuals with severe disease manifestations [29,31,32,33]. In contrast, encephalitic alphaviruses are symptomatic in only 5–10% of cases but can lead to serious neurological disease, with symptoms including headache, disorientation, seizures, and coma [34]. Mortality rates can range from <1% for VEEV to as high as 50–75% for EEEV among neuroinvasive cases [34,35].

Despite differences in acute presentations, a common feature among arthritogenic and encephalitic alphaviruses is that following resolution of **acute infection**, **long-term manifestations** are reported in up to 50% of affected individuals [34,36]. These include chronic arthralgia (e.g., in CHIKV [37,38,39,40], MAYV [41,42,43], RRV [44,45,46] and SINV [47,48]), fatigue (e.g., in CHIKV [49,50] and RRV [51]), and neurological sequelae (e.g., in EEEV [35], WEEV [52,53], VEEV [54] and CHIKV [55,56]). Additionally, emerging evidence indicates potential associations between alphavirus infections and long-term cardiovascular complications [57,58,59,60]. These **post-acute sequelae** raise the possibility that **persistent viral material** may contribute to these outcomes [61,62].

Alphavirus RNA and antigen have been detected in host tissues months after initial infection [63,64,65], challenging the long-held view that acute RNA viruses are fully cleared following resolution of acute symptoms [66]. This emerging paradigm is reinforced by evidence across several acute RNA viruses from different viral families—including Ebola virus [67,68,69], measles virus [70,71], Zika virus [72], Sendai virus [73], and SARS-CoV-2 [74]—which collectively demonstrate that viral RNA and/or viral antigens persist within host tissues for months to years. Persistent viral material has been implicated in chronic inflammation [61,73], post-acute sequelae [75,76,77], ongoing viral shedding [78], and even recurrence of infection [67,68]. Identification of the mechanisms underlying persistence of alphavirus infections is crucial in advancing our knowledge of how acute RNA viruses interact with the host cell and the implications for clinical outcomes, therapeutic development, and global health preparedness. 

In this review, we provide a comprehensive synthesis of the current evidence supporting alphavirus RNA persistence in vertebrates and highlight gaps in knowledge. Emerging findings suggest that alphaviruses may exhibit a broader tissue tropism than previously appreciated. However, limited access to **immune-privileged** tissues in humans may obscure our understanding of a more complex and widespread pattern of persistent infection. Integrating insights from *in vitro* and *in vivo* studies of alphavirus RNA persistence, we discuss the potential role of **viral persistence** in chronic disease and the viral and host determinants that may drive it. Uncovering the hidden mechanisms of viral persistence and their contribution to post-acute sequelae will shed light on this underrecognized global health threat.

## 2. Alphavirus Persistence: Insights from Human and Animal Models

Alphavirus infections of vertebrate cells are typically lytic and driven by host transcriptional and translational shutoff, leading to cytopathic effects and death of the infected cell (reviewed in [79]). The alphavirus +ssRNA genome is capped at the 5’ end and polyadenylated at the 3’ end, allowing direct translation by the host mRNA translation machinery. The genome is organized into two open reading frames (ORFs). The first ORF encodes the nonstructural proteins 1-4 (nsP1-4). These proteins form the viral replication complex, which is responsible for generating the negative-sense RNA intermediate and subsequently synthesizing both the full-length genomic and the 26S subgenomic RNAs. The second ORF is translated from the 26S subgenomic RNA and encodes the structural polyprotein. This precursor is processed into the capsid protein, the hydrophobic 6K accessory protein and its frameshift product TF, and the envelope glycoproteins E3, E2, and E1. These proteins are critical for virion assembly, budding, and host cell entry. Mature virions display 80 trimeric spikes on their surface, each composed of E1-E2 heterodimers that mediate receptor binding, endocytosis, and membrane fusion (reviewed in [80]).

These viruses establish infection at the site of the mosquito bite, targeting cells within the skin dermis, including fibroblasts, keratinocytes, melanocytes, and immune cells such as macrophages and Langerhans cells. From this initial site, the virus disseminates to the blood and draining lymph nodes (dLNs), where replication occurs before systemic spread [2]. Following acute infection, alphavirus RNA and proteins can persist for months to years after **viremia** resolves. The acute phase typically spans the first 1–2 weeks post-exposure and is characterized by high-level viral replication before immune responses drive resolution. In contrast, persistence refers to the continued presence of viral RNA or antigen for weeks to years beyond the acute phase, often at low levels, yet sufficient to sustain inflammation and contribute to chronic disease manifestations [66]. Whether this persistence reflects ongoing low-level viral replication, is due to a non-replicative remanent of viral RNA or antigen, or contributes to chronic disease, is an area of active investigation. Here, we present current evidence from human and animal models.

### 2.1. Arthritogenic Alphaviruses: Evidence of Viral Persistence in Humans and Non-Human Primates

Among arthritogenic alphaviruses, the strongest evidence for persistence in humans comes from CHIKV and RRV, for which viral RNA and antigens have been detected in human musculoskeletal tissue specimens months to years after acute infection [63,64] (Figure 1). In CHIKV-convalescent individuals, viral RNA and E2 protein were detected in perivascular synovial macrophages 18 months post-infection [64], and viral antigens were observed in skeletal muscle progenitor cells up to three months after symptom onset [81]. For RRV, viral RNA was found in knee biopsies five weeks after symptom onset, specifically within inflamed synovium [63]. Notably, CHIKV E1 antigen was detected in 14 of 17 joint biopsies collected 2–3 years post-infection, despite no detectable viral RNA [82], suggesting viral protein persistence or viral RNA levels below assay detection thresholds. In contrast, other studies examining synovial fluid from individuals with CHIKV-associated chronic arthralgia reported no detectable viral RNA or antigen, suggesting that, in some cases, chronic inflammation may reflect autoimmune mechanisms rather than persistent virus [83,84]. Whether chronic inflammation in joint tissue after alphavirus infection is driven by persistent viral material, immune dysregulation, or both remains unresolved. Beyond musculoskeletal tissues, CHIKV RNA has been detected in semen, urine, and saliva months after acute infection [85,86], and CHIKV RNA and E2 protein have been identified in the cervical tissues of two women up to six months post-infection [87]. Collectively, these observations suggest that alphavirus RNA and antigen may persist in a wider range of tissues than previously recognized, with potential implications for viral biology, immune responses, and long-term disease outcomes. These additional sites may contribute to tissue-specific pathology or even transmission potential. However, the prevalence and functional relevance of these putative reservoirs remain poorly defined.

In addition to the detection of viral RNA and antigen in biological specimens, serological studies raise the possibility of persistent antigen stimulation in infected individuals. In La Réunion, anti-CHIKV IgM antibodies were detected for up to 13 years post-infection, possibly reflecting ongoing antigen exposure [40]. Likewise, similar findings have been described during RRV [88] and SINV [89] epidemics. While the presence of long-lived IgM antibodies cannot be explained solely by ongoing viral antigen exposure [90], their prolonged detection is compatible with potential continued antigenic stimulation, providing an additional line of evidence supporting the hypothesis that alphaviruses can persist in humans after resolution of acute infection.

Evidence from non-human primate (NHP) models further demonstrates the ability of arthritogenic alphaviruses to persist beyond the acute phase. In cynomolgus macaques, CHIKV RNA and antigen persist for months in joint-associated tissues, the liver, and lymphoid organs, particularly within joint and splenic macrophages [91]. Interestingly, infectious particles were reported in the spleen and liver at 44 days post-infection (dpi), suggesting ongoing CHIKV replication [91]. However, this interpretation is limited by the absence of mock-infected controls in the TCID_50_ quantifications, which cannot rule out the contribution of tissue-derived toxicity to the observed cytopathic effect. Furthermore, comparative studies in NHPs revealed that aged macaques exhibit significantly greater and more prolonged viral replication than younger adults, with viral RNA detectable in the spleen at 35 dpi [92]. These findings suggest that the host immunological state may influence the propensity of viral RNA to persist and that naturally age-associated changes in innate and adaptive immunity may further increase the risk of viral persistence.

Collectively, these findings indicate that arthritogenic alphaviruses can persist in joint-associated, lymphoid, and reproductive tissues in humans. However, whether the detected viral RNA and antigen correspond to actively replicating virus, full-length infectious genomes, or residual **viral debris** remains to be defined.

**Figure 1 viruses-18-00030-f001:**
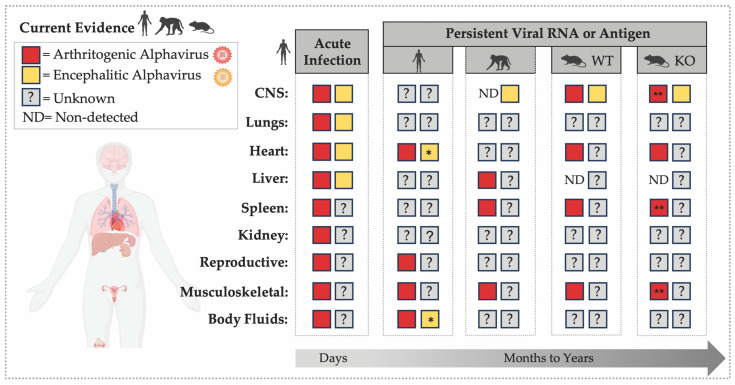
**RNA or Antigen Detection during Acute and Persistent Alphavirus Infection across Organs.** The figure highlights the temporal and tissue-specific distribution of alphavirus detection across vertebrate tissues, including the central nervous system (CNS) [52,53,65,91,93,94,95,96,97,98,99,100,101], lungs [95,100,102], heart [95,100,102,103], liver [91,95,100,102,104,105], spleen [91,92,95,100,104,106,107]; kidney [95,100], reproductive tissues [87], musculoskeletal tissues [63,64,81,82,91,104,106,107,108,109,110,111], and body fluids [85,86,87,102]. The left panel summarizes evidence of acute alphavirus infection (days post-infection) in humans, illustrating detection of viral RNA or antigen across multiple organs and body fluids (semen, urine, saliva, whole blood). The right panels summarize published evidence of alphavirus RNA or antigen persistence (months to years) in humans, non-human primates (NHPs), and mouse models, including immunocompetent (WT) and immunodeficient (KO) strains such as *Rag1*^−/−^, *μMT*, *scid*, and *Mavs*^−/−^. An asterisk (*) indicates findings from organ transplantation cases in which individuals were immunosuppressed during infection. Double asterisks (**) indicate detection of infectious virus in organs up to 40 dpi in *Rag*^−/−^ mice.

### 2.2. Encephalitic Alphaviruses: Evidence of Viral Persistence in Humans and NHP Models

In contrast to arthritogenic alphaviruses, which have several documented cases of viral RNA or antigen detection months after infection in humans, evidence of viral persistence following encephalitic alphavirus infection remains limited. These infections are frequently associated with chronic neurological sequelae, including cognitive, motor, behavioral, and sensory impairments, typically affecting infants and children, older adults, and those with immunological deficiencies [34,35]. The possibility that these long-term neurological outcomes may be linked to persistent viral infection was first proposed in the 1940s by Noran and Baker [52,65]. However, only a few studies to date have detected viral RNA or infectious virus in human tissues, and most derive from the acute phase of infection (Figure 1). For example, VEEV RNA was detected in the brain tissue of an infant who died of infection [97], and WEEV was isolated from the brains of several infected individuals [52,53].

Whether neurological sequelae are driven by residual viral RNA and/or antigen in the central nervous system (CNS) remains unresolved, in part because access to human brain or spinal cord tissue is typically limited to post-mortem sampling. Unlike synovial biopsies, in which persistent viral RNA and antigen have been detected, CNS samples are rarely available for research. This limitation hinders efforts to identify long-term viral reservoirs or determine whether viral RNA, protein, or antigens persist in these immune-privileged tissues. Consequently, NHP models have become indispensable for investigating encephalitic alphavirus persistence. In surviving NHPs, viral RNA has been detected in the brain several weeks after infection with WEEV, EEEV, or VEEV, persisting beyond the resolution of viremia and in the absence of detectable infectious particles for at least one month after infection [93,94,96]. For instance, cynomolgus macaques exposed to VEEV that developed neurological sequelae retained viral RNA in the brain for up to 52 dpi and exhibited CNS inflammation [93], supporting the hypothesis that alphavirus-mediated neurological sequelae may be partially driven by persisting viral infection and/or residual viral RNA or antigen. Collectively, these animal studies provide compelling evidence that encephalitic alphaviruses can persist in the CNS, where viral RNA or antigens may sustain chronic inflammation and contribute to neurological sequelae. However, whether a direct causal relationship exists between persistent viral RNA and these long-term neurological sequelae remains to be determined.

### 2.3. From Correlation to Causation: Mouse Models as Key Tools to Study Alphavirus Persistence

A central unresolved question is whether persistent alphavirus RNA reflects ongoing viral replication and whether this persistence causally contributes to the chronic arthritogenic or encephalitic sequelae observed in humans. Because establishing causality requires experimental systems amenable to targeted perturbation, mouse models offer a powerful platform to dissect the mechanisms of viral persistence within the context of an intact immune system.

Immunocompetent mouse models of alphavirus-induced arthralgia recapitulate key features of human disease, including long-term detection of viral RNA in joint-associated tissues and the spleen [104,106,108,112]. Joint fibroblasts, skeletal myofibers, macrophages, and splenic lymphoid cells have been identified as major cellular reservoirs of alphavirus RNA months after infection [107,109,110]. However, whether this RNA represents replication-competent genomes or residual, non-replicative fragments remains under debate [109,113,114]. The detection of positive- and negative-sense CHIKV RNA in joints up to 100 dpi, together with persistent antigen and sustained type I interferon (IFN-I) and T-cell responses, supports the possibility of ongoing active replication [115,116]. Consistent with this, immunosuppression or antiviral treatment during the chronic stage of disease has revealed evidence of active viral RNA replication one month after infection for both RRV [113] and CHIKV [109], with one study additionally reporting the isolation of infectious CHIKV at 90 dpi [112]. Moreover, a recent preprint examining CHIKV infection in joint-associated tissues at single-cell resolution identified actively replicating CHIKV RNA in fibroblasts and macrophages at 28 dpi [109], further supporting the idea that alphaviruses can maintain long-lasting replication in specific **cellular niches**.

The increased severity of alphavirus-associated pathologies in young children, older adults, and individuals with comorbidities underscores the role of host immunity in shaping disease outcomes. Similar to observations in NHPs, aged mice exhibit higher levels of CHIKV RNA in joint tissues at 60 dpi compared to young adults, a difference attributed in part to elevated transforming growth factor beta (TGF-β) levels that impair neutralizing antibody production [112]. Consistent with the importance of adaptive immunity in controlling viral burden, mice lacking B and/or T cells develop persistent viremia, markedly elevated tissue RNA loads, pronounced inflammation, and detectable viral RNA for ≥16 weeks [104,115]. Robust adaptive immune responses limit MAYV persistence [105], highlighting a conserved role for B- and T-cell responses across arthritogenic alphaviruses. These mechanisms are further discussed in Section 5.

Mouse models of encephalitic alphavirus infection have been instrumental in dissecting the causative drivers of neurotropism and pathogenesis. Studies using SINV, an arthritogenic alphavirus phylogenetically related to WEEV and widely employed as a model of alphavirus-induced encephalitis in mice, and SFV demonstrated that alphavirus RNA can persist in the brain in full-length infectious form for months to years [98]. Following immunosuppression, viral RNA can resume active replication [98,99]. Notably, administration of antibody-mediated immunosuppression to immunocompetent mice 18 months after SFV infection reactivated **productive infection**, with infectious particles detected in the brain [98]. These findings show that neurotropic alphaviruses can persist long term in the CNS in their full-length infectious form, with the immune system playing a central role in suppressing reactivation and limiting relapse. However, whether this persistence underlies chronic neurological sequelae remains unknown.

Taken together, available evidence indicates that alphaviruses can persist across multiple organ systems (Figure 1), with outcomes shaped by the host immune landscape. Older adults appear particularly susceptible to impaired viral clearance, potentially creating a permissive environment for alphavirus persistence. Current data support several non-mutually exclusive scenarios: (1) alphaviruses establish low-level replication within specific cellular niches; (2) lingering, non-replicative RNA or antigen may drive chronic inflammation; and (3) viral RNA detected in macrophages may reflect phagocytic uptake rather than productive infection. A crucial next step is to define the molecular and immunological mechanisms that sustain alphavirus persistence.

## 3. The Establishment of Persistent Infections: Finding the Right Place to Hide

A central question in alphavirus persistence is whether specific cell types or tissues harbor intrinsic features that allow them to survive acute infection and retain viral RNA and/or antigen long term. Recent post-mortem studies suggest that acute alphavirus infection in humans may involve a broader tissue tropism than previously appreciated. In fatal CHIKV cases, viral RNA was detected not only in joint-associated tissues but also in the cerebrospinal fluid (100%), spleen (52%), lungs (44%), liver (28%), heart (20%), kidney (20%), and brain (13%) [100]. In another study, CHIKV antigen was found in the blood vessels, heart, skin, skeletal muscle, and visceral tissues in over 75% of examined autopsies [95]. Similarly, among encephalitic alphaviruses, infectious EEEV was detected in non-neural tissues, including the heart, liver, and lungs, in a case of transplant-associated EEEV transmission that resulted in encephalitis in all the organ recipients [102]. Remarkably, longitudinal endomyocardial biopsies from the heart transplant recipient revealed persistent EEEV RNA and antigen up to 96 days post-transplantation [102]. While intriguing, immunosuppression in the transplant recipients likely altered viral immune clearance dynamics. Together, these findings both reinforce that alphaviruses disseminate across multiple tissues and raise the possibility that they may establish persistence beyond primary sites of infection (Figure 1). In this section, we outline how alphaviruses engage specific cellular niches that may support long-term persistence.

### 3.1. Viral Entry: Receptor-Dependent and Receptor-Independent Entry

Given the similarities in receptor usage among alphaviruses [117,118], it can be speculated that dissemination to multiple organs beyond the primary infection site may represent a shared feature across evolutionarily related alphaviruses. Alphaviruses utilize a variety of host cell surface receptors to mediate viral attachment and entry (reviewed in [119]). The viral envelope glycoproteins E1 and E2 function cooperatively during this process. E2 acts as the attachment protein, binding host cell receptors and entry factors, while E1—a class II fusion protein—mediates membrane fusion within the endosome [80]. The expression of these receptors across diverse tissues may contribute to the broad tissue tropism observed during infection. For arthritogenic alphaviruses, the MXRA8 (matrix remodeling associated 8) protein has emerged as a well-characterized entry factor [117]. MXRA8 is broadly expressed on epithelial, endothelial, and mesenchymal cells across diverse tissues, including the joints, muscles, heart, liver, and brain [117,120]. Additionally, CHIKV can also bind CD147, phosphatidylserine receptors such as TIM-1 (T-cell immunoglobulin mucin domain-1), and prohibitin-1, suggesting alternative entry routes in cell types or tissues with no MXRA8 expression [119].

Interestingly, the arthritogenic SINV does not use MXRA8. Instead, SINV and encephalitic alphaviruses appear to exploit distinct receptors highly expressed in the CNS. For instance, LDLRAD3 (low density lipoprotein receptor class A domain-containing protein 3) was recently identified as a receptor for VEEV [121], while PCDH10 (protocadherin-10) was found to serve as a human receptor targeted by WEEV [122]. Moreover, related lipoprotein receptors, including VLDLR (very low density lipoprotein receptor) and ApoER2 (low density lipoprotein receptor-related protein 8), can mediate infection by SINV, SFV, and EEEV [118]. The diversity and broad distribution of these receptors underlie alphaviruses’ ability to infect multiple organ systems, including the brain, heart, liver, and spleen, challenging the traditional view of alphaviruses as strictly arthritogenic or neurotropic.

Although receptor-mediated entry is crucial for initial infection, alternative mechanisms of viral spread, such as cell-to-cell transfer and trafficking within immune cells or extracellular vesicles, are now recognized as important contributors to viral dissemination. Recent studies demonstrated that CHIKV can disseminate via receptor-independent mechanisms involving the formation of intracellular long extensions, potentially enabling viral spread to cells lacking canonical receptors while evading neutralizing antibodies [123]. Further, trafficking via viral-RNA-containing extracellular vesicles [124] and virus-associated immune cells [125] may contribute to viral dissemination within the infected host. These mechanisms could enable the virus to expand its tropism, reaching a broader range of cells and tissues. This expanded perspective is critical for understanding alphavirus pathogenesis, potential reservoirs of persistence, and the systemic manifestations observed during alphavirus infections. Within this framework, a central question emerges: what makes a given cell type uniquely capable of becoming a reservoir for alphavirus persistence?

### 3.2. Cell Types Implicated in Alphavirus Persistence

How alphavirus RNA persists within a host for months to years remains an open question. One possibility is that persistence arises in long-lived cells that survive the initial infection. In that scenario, viral RNA may persist as a full-length genome with minimal to no replication or as remnant RNA fragments from the initial infection. Alternatively, persistence may reflect low-level replication with limited spread to neighboring cells or uptake by phagocytic cells through either direct infection or engulfment of infected material during clearance. Several cell types, including macrophages, neurons, and dermal and muscle fibroblasts, have emerged as candidate reservoirs capable of sustaining alphavirus RNA long after acute infection.

#### 3.2.1. Tissue-Resident and Monocyte-Derived Macrophages

Macrophages are key effectors of host innate immunity and comprise embryonically derived, self-renewing tissue-resident populations (e.g., microglia, Kupffer cells, and CX3CR1^+^ synovial macrophages) as well as short-lived, monocyte-derived macrophages (MDMs) replenished from circulating monocytes [126,127]. Clinical and experimental studies have identified macrophages as major targets of alphavirus infection and persistence, promoting infection-driven chronic inflammation [64,91,109,128]. CHIKV RNA has been detected in CD14^+^ macrophages from synovial tissue years after infection and has been implicated in chronic inflammation and arthralgia [64]. During RRV infection, synovial macrophages are likely to be the main reservoir for viral RNA detected in biopsies [128]. In NHPs, CD68^+^ splenic macrophages retain CHIKV RNA up to 90 dpi [91], and fatal CHIKV cases show viral antigen within macrophages of the spleen and tendon, and in sinusoidal Kupffer cells [95]. Thus, macrophages not only contribute to viral clearance but also may sustain infection and persistence.

Due to these cells’ strong phagocytic capacity, distinguishing true infection from uptake of viral material is challenging *in vivo*. However, *in vitro*, encephalitic alphaviruses such as VEEV can productively replicate in skin-resident macrophages and microglia [129,130]. MDMs—in which replication of other viruses such as SARS-CoV-2 and some strains of influenza A, is largely **abortive** [131,132,133]—can sustain productive replication of several arthritogenic alphaviruses, like RRV and CHIKV [134,135]. For example, CHIKV generates infectious particles (plaque-forming units, or PFU) in MDMs, with infection occurring in only 5–50% of human primary cells or ~1% of the THP-1 cell line [134,136]. Remarkably, infection of even 1% of cells yields moderate to high viral loads (10^4^–10^6^ PFU/mL) [134,136], underscoring potential functional heterogeneity within macrophage populations. Mechanistically, CHIKV infection is restricted by **host factors** acting at late stages of the viral life cycle. For instance, the signal peptidase complex subunit 3 (SPCS3) and eukaryotic translation initiation factor 3 subunit K (eIF3k) inhibit viral budding [134]. Evidence of positive selection within the CHIKV E1 glycoprotein suggests that it interacts with these host factors at evolutionarily selected sites, highlighting E1 as a viral antagonist that counteracts their antiviral activity [134].

These findings support the idea that CHIKV, and potentially other alphaviruses, can antagonize host restriction factors to enable productive infection and persistence within macrophages. In line with this, a recent study demonstrated that macrophages are the predominant cell type harboring replicating CHIKV, MAYV, and RRV RNA in joint tissues of immunocompetent mice at 28 dpi [109] and showed that the presence of replicating persisting virus in macrophages is linked to joint inflammation. Taken together, these studies reveal that productive infection and viral antagonism can shape alphavirus replication and persistence within macrophages, although whether these processes differ between macrophage lineages remains to be defined.

#### 3.2.2. Neurons and Non-Neuronal CNS Cells

Neurons are highly plastic, long-lived, terminally differentiated, and relatively immune-privileged cells capable of continuous structural and functional remodeling throughout the entire lifespan of the individual [137,138]. Neurons have been implicated as reservoirs of alphavirus RNA in animal models. For instance, SINV RNA has been detected in mouse brains up to 17 months post-infection, localized within neurons even in the absence of infectious virus [99]. *In vitro* studies further demonstrate that neuronal maturation influences the establishment of SINV persistence, underscoring the importance of the cellular differentiation state in determining infection outcomes [139]. Similarly, SFV RNA has been shown to persist in neurons for over a year in the absence of detectable infectious virus, with productive infection re-appearing following immunosuppression [98]. The low expression or absence of surface major histocompatibility complex class I (MHC-I) on neurons contributes to their immune-privileged status, potentially facilitating viral evasion and long-term persistence.

Non-neuronal CNS cells such as oligodendrocytes and astrocytes are also long lived and can be infected by arthritogenic alphaviruses both *in vitro* and *in vivo* [140,141,142]. The detection of persistent SFV antigen within astrocytes in infected immunodeficient mice further highlights the potential for these glial cell populations to contribute to alphavirus persistence within the CNS [141].

#### 3.2.3. Fibroblasts and Muscle Satellite Cells

Musculoskeletal cell types have been identified as potential reservoirs for arthritogenic alphaviruses. Specifically, fibroblasts are long-lived stromal cells with functional plasticity that act as immune regulators, displaying both pro-inflammatory and immunosuppressive properties during chronic insults [143]. Dermal and muscle fibroblasts, as well as skeletal myofibers, have been shown to harbor CHIKV RNA in an immunocompetent host for up to 28 dpi in the absence of detectable infectious virus [109,110], with the presence of negative-sense RNA supporting active replication [109]. Human muscle satellite cells, stem-like progenitors essential for cellular regeneration, also appear to be particularly susceptible to CHIKV persistence [81]. These findings highlight fibroblast and satellite cells as sites of alphavirus persistence in musculoskeletal tissue. However, how infection in fibroblasts could result in a non-lytic persistent state remains to be defined. Recent studies have demonstrated that arthritogenic alphaviruses can evade cytotoxic T lymphocyte (CD8^+^ T-cell) recognition *in vivo* by downregulating MHC-I antigen presentation [144]. This suggests that, rather than fibroblasts being inherently immune-privileged reservoirs, viruses can fine-tune cellular responses to render themselves refractory to immune recognition.

#### 3.2.4. Splenic Lymphoid Cells

Splenic lymphoid cells include follicular dendritic cells (FDCs), germinal center (GC) B cells, T cells, plasma cells, monocytes, and macrophages, which together form a dynamic immunological network [145]. Within the GC, FDCs and B cells occupy an immune-privileged environment, where FDCs retain **opsonized antigens** and present immune complexes through complement and Fc receptors for extended periods—ranging from months to years—while GC B cells differentiate into long-lived plasma and memory cells [145]. Several studies have detected alphavirus RNA in the spleens of infected NHPs and mice for months post-infection, implicating this tissue as a site of persistence [91,92]. In CHIKV-infected NHPs, viral RNA was observed in splenic macrophages at 19 dpi, and viral antigen persisted in mononuclear cells infiltrating B-cell zones up to 90 dpi [91]. Similarly, in mice, replicating viral RNA localized within splenic GC B cells, FDCs, and CD11c^+^ cells at 14 dpi [107]. These findings suggest that multiple splenic immune cell subsets, beyond macrophages alone, may harbor persistent alphavirus RNA. However, the long-term impact of this reservoir on the overall immune response against alphaviruses remains unclear.

Taken together, these findings suggest that alphavirus persistence may arise from a complex interplay among viral entry strategies, receptor usage, host permissiveness, and mechanisms of dissemination. Evidence supports that alphaviruses can establish persistent infections across multiple cellular environments, potentially contributing to the wide range of chronic pathologies beyond the classic joint pain or neurological sequelae (including fatigue and depression, among others). These potential reservoirs share common features—longevity, plasticity, and relative immune privilege—that may enable them to harbor viral RNA for weeks, months, or even years.

## 4. Virus–Host Interactions in Alphavirus Persistence: Shifting from Lytic to Non-Lytic Infection

The alphavirus life cycle in vertebrate cells typically results in lytic infections and involves viral entry, cytoplasmic replication and translation of viral RNA, and assembly and budding of progeny virions at the plasma membrane [2]. To establish persistence within a vertebrate host, however, alphaviruses must reprogram host pathways to transition from a productive lytic infection to a non-lytic or low-replicative state compatible with long-term cell survival. Lytic infection is driven largely by alphavirus-induced transcriptional and translational shutoff, which triggers cytopathic effects and cell death (reviewed in [79]). For example, SINV nsP2 mediates host transcriptional shutoff [146], while VEEV relies on its capsid protein for the same function [147]. Pioneering *in vitro* studies using replicon systems—later validated with full-length viruses—demonstrated that both encephalitic and arthritogenic alphaviruses can establish non-cytopathic replication in mammalian cells through adaptive mutations that reduce cytopathicity while maintaining replication [148,149,150,151,152]. Mutations in SINV nsP2 [151,152], CHIKV nsP3 [150], and VEEV capsid protein [149] enable persistent, non-cytopathic RNA replication in otherwise susceptible cell lines for weeks to months. Collectively, these studies provided the first mechanistic evidence that alphaviruses possess an intrinsic capacity to establish persistent infection in vertebrate cells.

Based on *in vitro* and *in vivo* evidence, three non-mutually exclusive mechanisms have emerged as potential contributors to alphavirus persistence: (1) generation of defective viral genomes (DVGs) and defective interfering particles (DIs); (2) antagonism of apoptosis; and (3) cellular permissiveness shaped by host factors (Figure 2). Cellular permissiveness, influenced by the expression of proviral or antiviral genes, determines whether viral replication proceeds productively, abortively, or in a restricted manner. In the following section, we examine each mechanism and its potential contributions to alphavirus persistence.

### 4.1. Limitation of Replication Output and Cytopathicity Through DVGs and DIs

One well-established mechanism contributing to RNA persistence is the generation of DVGs and DIs (reviewed in [153]). DVGs arise during error-prone viral replication and, when encapsidated, form DIs. Lacking essential coding regions, these genomes can only replicate in the presence of a **helper virus**. DVGs and DIs are thought to promote persistence either by competing with full-length genomes for viral and host resources [154] or by inducing pro-survival cellular states [153].

Foundational alphavirus studies support a role for DIs in shaping persistent infection. During SINV infection, DIs contributed to persistent infection in baby hamster kidney (BHK-21) cells for over 100 days [155,156]. In SFV infection, co-inoculation of mice with DIs converted an otherwise lethal challenge into long-term, clinically silent viral persistence, with infectious virus detectable for up to 6.5 months [157]. More recently, CHIKV-derived DVGs were detected in mammalian cell culture and in mosquitoes, where they limited replication *in vitro* and reduced dissemination *in vivo* [158]. Mechanistically, *in vitro* studies using SINV identified **low-fidelity variants of the RNA polymerase nsP4** as a driver of DI formation through elevated error rates and recombination events [159].

Although DVGs and DIs have not yet been documented in natural human alphavirus infections, they are well described for other RNA viruses, including hepatitis C and respiratory syncytial virus [153], highlighting their potential contribution to alphavirus persistence. In Sendai virus infection, DVGs trigger a mitochondrial antiviral signaling protein (MAVS)-dependent tumor necrosis factor (TNF) response that eliminates highly infected cells while preserving DVG-rich cells via TNFR2/TRAF1 pro-survival pathways, creating a long-lived infected cell population that supports persistent infection [160]. These findings illustrate how DVGs can reprogram host responses to favor cell survival during viral infection. Whether similar processes occur during alphavirus infections in humans remains unknown, but these *in vitro* and *in vivo* findings suggest that by lowering viral burden, dampening cytopathic effects, and modulating immune pathways, DVGs and DIs may play a central role in alphavirus persistence.

### 4.2. Antagonism of Apoptosis

Productive alphavirus infection in vertebrates typically triggers programmed cell death, most often via apoptosis, though necroptosis and pyroptosis may also occur [161]. Apoptosis is initiated by activation of pro-apoptotic proteins such as BCL-2-like protein 4 (BAX), leading to mitochondrial permeabilization, cytochrome c release, and caspase activation, which break down the cell into apoptotic bodies cleared by immune cells [162]. For persistence to be established, these pathways must be suppressed, delayed, or repurposed to allow long-term cell survival. Early SINV studies demonstrated that neurons resist virus-induced apoptosis, in part through expression of the anti-apoptotic protein B-cell lymphoma 2 (BCL-2), enabling long-term maintenance of SINV RNA [163]. Overexpression of BCL-2 in AT-3 mouse mammary carcinoma cells leads to inhibition of virus-driven apoptosis allowing SINV RNA persistence for up to five months [163]. Comparable mechanisms are observed for other RNA viruses, such as hantaviruses, which upregulate BCL-2 in infected cells, block caspase activation and promote cell survival [164]. Alphaviruses have evolved strategies to modulate host cell death pathways in specific cell types. For instance, during RRV infection of human monocytic MM6 cells, apoptosis occurred in a subset of cells despite minimal viral replication, while a surviving fraction maintained detectable viral RNA for up to 40 dpi [165]. These divergent outcomes suggest that cell-intrinsic heterogeneity can permit the survival of subpopulations that retain viral RNA, creating potential niches of persistence.

Additional host processes may also shape persistence. **Autophagy**, for example, has been shown to limit apoptosis during CHIKV infection *in vitro* and *in vivo* [166,167], potentially enabling persistent niches. Conversely, apoptosis itself may indirectly promote dissemination and persistence, as recently shown for CHIKV. CHIKV can be packaged within apoptotic blebs, which can be engulfed by neighboring cells, including macrophages, thereby promoting viral dissemination while avoiding immune detection [168]. Together, these findings highlight that both host- and virus-mediated modulation of cellular death may contribute to the establishment and maintenance of alphavirus persistence.

### 4.3. Host Factors

Cellular permissiveness to alphavirus infections, defined as the ability of a cell to support the complete viral life cycle, is shaped by numerous host factors (reviewed in [169]). One factor, ribonuclease L (RNase L), an interferon-stimulated gene (ISG), has been directly implicated in regulating alphavirus persistence. RNase L normally restricts viral infection by degrading viral and host RNAs, leading to translational arrest and apoptosis [170,171]. Loss of RNase L promotes alphavirus persistence in mouse embryonic fibroblasts (MEFs), allowing cells to survive infection and maintain infectious virus for several passages [172]. In RNase-L-deficient MEFs, both SINV and SFV established persistence, likely due to sustained viral RNA replication via continued formation of replication complexes and avoidance of virus-driven apoptosis [172]. These findings further support evasion of apoptosis as a central mechanism facilitating alphavirus persistence.

Collectively, available evidence suggests that alphavirus persistence does not require high-level viral replication but instead relies on reprogramming host cellular processes to promote cell survival while evading immune-mediated clearance. Specifically, persistence in vertebrate cells may require active modulation of host processes that would otherwise trigger cell death and viral clearance. Current evidence points to several strategies by which alphaviruses may promote non-cytopathic infections, including limiting **replication output**, interfering with apoptosis, and altering host immune responses (Figure 2).

## 5. Establishing Persistence Within the Host: The Role of Immune Modulation

The innate immune response, particularly IFN-I signaling, constitutes the first line of antiviral defense, while adaptive immunity, mediated by virus-specific T and B lymphocytes, is essential for viral clearance and long-term protection [173]. Under normal circumstances, these mechanisms cooperate to eliminate infection. For persistence to arise, however, alphaviruses must evade these responses, creating a permissive environment in which viral RNA and antigen can be maintained long after acute infection has resolved. Current evidence indicates that modulation of IFN-I signaling, impairment of antigen presentation, and remodeling of local immune microenvironments can promote persistence despite active immune surveillance. Although many of these processes remain incompletely understood, recent studies have begun to reveal key mechanisms by which alphaviruses might modulate host immunity to favor long-term infection.

### 5.1. IFN-I and Innate Control of Alphavirus Infection

IFN-I signaling is a critical host determinant of alphavirus control (reviewed in [169]). The requirement of IFN-I signaling to limit alphavirus pathogenesis has been demonstrated for both arthritogenic and encephalitic alphaviruses, with mice lacking the interferon alpha/beta receptor (IFNAR1) in non-hematopoietic cells succumbing to infection, even with attenuated alphavirus strains [174,175,176,177]. During infection, alphavirus RNAs are sensed by pattern-recognition receptors (PRRs), including cytosolic retinoic acid-inducible gene-I (RIG-I) and melanoma differentiation-associated protein 5 (MDA5), and endosomal Toll-like receptors 3 (TLR3) and 7 (TLR7). These pathways signal through MAVS, TIR-domain-containing adapter-inducing interferon beta (TRIF), and myeloid differentiation primary response 88 (MyD88), respectively, to induce IFN-I, type III interferon (IFN-III), and pro-inflammatory cytokines [2]. IFN-I then acts in an autocrine and paracrine manner via IFNAR1, triggering JAK-STAT signaling (JAK, Janus kinase; STAT, signal transducers and activators of transcription) and inducing the expression of hundreds of ISGs that restrict viral infection at multiple stages of the replication cycle. The impact of IFN-I signaling on infection outcomes is highly context dependent, varying with the timing of induction, magnitude of the downstream response, tissue type, cell type, and pathogen (reviewed in [178]).

A recent study using mouse models revealed a cell-type-specific dependence on interferon alpha (IFNα) in restricting CHIKV persistence. In this work, the authors showed that IFNα signaling within the first 48 h of infection regulates long-term cell survival at sites of viral dissemination [111]. This early IFNα-driven response promoted enhanced persistence within fibroblasts and immune cells for at least 28 dpi within joint tissue, whereas myofibers were not affected [111], indicating a cell-type-specific dependency on IFNα and revealing surprising complexity in IFN-I-mediated control of persistence. Additionally, we have recently shown that efficient IFN-I responses in cardiac cells are essential for CHIKV clearance, whereas disruption of IFN-I signaling through MAVS deficiency leads to persistent viral RNA in cardiac tissue and chronic inflammation of the heart and the aorta [103].

Interestingly, host age can profoundly influence IFN-I responses. Neutralizing IFN-I autoantibodies—detected in approximately 4% of individuals over 70 years of age [179]—have been implicated in severe RRV disease [180] and may contribute to impaired viral clearance in the elderly population. Consistent with this, studies using NHPs or mouse models revealed that aged animals exhibited significantly greater and more prolonged viral replication than younger adults, with viral RNA detectable in tissues for months [92,112]. These findings align with broader principles of immunosenescence, whereby age-related changes in innate sensing, interferon signaling, and T-cell function compromise antiviral defense [181], potentially creating permissive conditions for viral persistence.

In addition to host-driven modulation, alphaviruses can antagonize IFN-I responses and favor persistent phenotypes. For example, RRV isolated from persistently infected RAW 264.7 macrophages acquired enhanced resistance to IFN-I, a phenotype associated with increased pathogenicity *in vivo* [182]. These findings suggest that selective pressures during chronic infection may drive the emergence of viral variants with reduced IFN-I sensitivity.

Collectively, these findings underscore the central role of IFN-I signaling in balancing viral clearance and persistence. The timing, magnitude, and cellular context of IFN-I signaling may each play distinct roles in determining infection outcomes, and understanding these virus-specific host dynamics remains a key priority for elucidating their contribution to alphavirus persistence. Further studies focusing on persistent states are needed to delineate how distinct IFN subtypes and tissue microenvironments regulate this balance.

### 5.2. Adaptive Immunity in Alphavirus Persistence

While innate immunity restricts early infection, adaptive immunity is responsible for pathogen-specific clearance and long-term protection. CD8^+^ T cells recognize viral peptides presented on MHC-I and eliminate infected cells across a broad range of tissues, whereas T helper lymphocytes (CD4^+^ T cells) recognize peptides presented on major histocompatibility complex class II (MHC-II), orchestrating cytokine production, B-cell activation, and CD8^+^ T-cell responses [2]. Activated B cells produce neutralizing antibodies that facilitate clearance through opsonization and complement activation. Despite this robust antiviral network, alphaviruses can persist by employing multiple strategies to evade or dampen adaptive immunity, prolonging their survival within the host [2].

Adaptive immunity is essential for the clearance of arthritogenic alphaviruses [104,105]. Mice lacking B and T cells, such as those deficient in recombination activating gene 1 (*Rag1^–^*^/*–*^), exhibit significantly higher CHIKV and MAYV RNA levels, with infectious virus persisting in the circulation for several months compared to their immunocompetent counterparts [104,105]. Similarly, CHIKV infection in B-cell-deficient mice (μMT) or *Rag1^–^*^/*–*^ mice results in persistent viremia, lasting up to a year [115]. B cells in lymphoid tissue have been implicated as potential reservoirs of persistent CHIKV RNA [107], and consistently, μMT mice show minimal RNA in the spleen compared to immunocompetent controls [107], suggesting that B cells may both promote antibody-mediated clearance and sustain viral RNA within immune cell niches. In addition, activation of TNF receptor superfamily member 9 (CD137)—a co-stimulatory receptor expressed on T, natural killer (NK), and B cells that amplifies immune responses—by an agonistic antibody accelerates CHIKV RNA clearance in lymphoid tissues but not joint-associated tissues [107], underscoring the tissue-specific nature of viral reservoirs. Beyond host factors, virus-induced immune evasion can also contribute to persistence, as shown for a pathogenic CHIKV strain that evades E2-specific antibodies to establish persistence despite robust humoral responses [183].

For neurotropic alphaviruses such as SINV, viral RNA can persist in CNS neurons for up to 17 months despite ongoing antibody and T-cell responses [99,184]. Similarly, SFV RNA has been shown to persist in the brains of immunocompetent mice for 18 months and can be reactivated when antibody responses are compromised [98]. This suggests that viral clearance from neurons occurs via non-cytolytic, antibody-mediated mechanisms that restrict rather than fully eliminate viral replication. In VEEV infection, persistence of viral RNA for up to 92 dpi in the brains of T-cell-deficient mice underscores the critical requirement of T-cell-mediated mechanisms for viral RNA clearance [101]. These findings highlight the crucial role of adaptive immunity in promoting viral clearance and provide evidence that alphavirus persistence is antibody modulated and tissue restricted, emphasizing the need to define all viral reservoirs of persistence within the host.

Another important component of adaptive immunity is T-cell-mediated cytotoxic clearance. While alphaviruses in neurons can evade CD8^+^ T-cell surveillance due to low expression levels of MHC-I, infection of other cell types requires alternative immune evasion mechanisms. Interestingly, during CHIKV infection, nsP2 impairs MHC-I trafficking to the plasma membrane, preventing peptide-loaded complexes from reaching the cell surface and enabling infected fibroblasts in joint tissue to avoid CD8^+^ T-cell-mediated killing and harbor persistent viral RNA [144]. Beyond direct immune evasion, alphaviruses can remodel lymphoid tissue architecture to undermine antibody-mediated control [185,186]. CHIKV triggers a rapid MyD88-dependent influx of monocytes and neutrophils into the dLNs within 24 h post-infection. These cells express inducible nitric oxide synthase (iNOS) and NADPH oxidase 2 (NOX2) through interferon regulatory factor 5 (IRF5) and IFNAR1 signaling, disrupting the structural organization of the dLNs [186]. This early inflammatory wave causes a loss of B- and T-cell compartment boundaries, preventing accumulation of naïve lymphocytes and antibody generation. Depletion of myeloid cells restores lymph node organization, increases plasma cell numbers, and enhances neutralizing antibody titers by 28 dpi [186]. In parallel, CHIKV suppresses high-endothelial-venule-mediated lymphocyte trafficking into draining lymph nodes and CCL21 upregulation, further delaying humoral responses [185]. This structural disruption during the acute phase likely weakens early adaptive priming and creates conditions conducive to viral RNA persistence in peripheral tissue.

### 5.3. Persistence as a Potential Driver of Chronic Pathogenesis

Adaptive immunity is required, yet often insufficient, for complete clearance of alphaviruses. Even in the presence of robust humoral and cellular responses, viral RNA and antigen can persist in select tissues, including joint, lymphoid, muscle, reproductive, and CNS compartments. Such persistence reflects a finely balanced interplay between viral immune evasion, tissue-specific immune privilege, and dysregulated cytokine activity. In this context, immune activation and viral immune escape coexist, illustrating how pathways that normally restrict viral replication may, when dysregulated, sustain chronic inflammation (Figure 3). Although direct causal evidence linking persistent alphavirus RNA to long-term pathologies remains lacking, a growing body of experimental and clinical observations suggests that viral persistence may maintain inflammatory programs that contribute to chronic disease [75]. Consistent with this concept, studies of respiratory paramyxovirus infection have demonstrated that long-term expression of viral antigens by surviving cells sustains innate immune activation and drives chronic pathogenesis, even in the presence of adaptive immunity [73]. Supporting a similar mechanism in alphavirus infection, a recent preprint examining CHIKV-infected mouse joints showed that treatment with a small-molecule inhibitor of viral replication reduced both CHIKV RNA levels and inflammatory gene signatures [109]. These findings provide the first experimental indication that persistent alphavirus RNA arising from ongoing low-level replication may contribute to chronic inflammation. Therapeutically, this duality highlights the need for strategies that promote effective adaptive viral clearance while minimizing the risk of exacerbating inflammatory damage.

## 6. The Need for Advanced Human Models to Study Alphavirus Persistence

A major barrier to studying the mechanisms underlying alphavirus persistence is the lack of experimental models that faithfully recapitulate the cellular diversity, spatial organization, and longevity of human tissues. Existing *in vitro* systems are limited by their short lifespan, reduced complexity, and incomplete capture of tissue-specific phenotypes. For example, macrophage models commonly used to study alphavirus infection rely mainly on monocyte-derived cell lines (e.g., THP-1) or primary human monocytes, which only partially mimic long-lived tissue-resident macrophages [134,136]. There is a critical need for human platforms capable of sustaining long-term, multicellular interactions that reflect the *in vivo* environments where persistent infection and chronic inflammation arise. Recent progress using primary human cells and induced pluripotent stem cells (iPSCs) have begun to address this gap. iPSC-derived organoids are now integrating diverse immune and stromal components, such as colonic organoids with resident macrophages [187] and cardiac organoids containing vessel-associated macrophages [188]. Extending these systems to include microglial, skin-resident, and synovial macrophages would provide complex physiologically relevant human systems to interrogate viral persistence in tissues targeted by arthritogenic and encephalitic alphaviruses. Human tissue explants also provide valuable models to examine viral and host responses within intact tissue architecture [133,189]; however, the limited lifespan of current explant cultures constrains their use for studying long-term persistent infections.

Beyond macrophages, human primary cells remain valuable platforms for studying viral persistence. However, despite their physiological relevance, these models are limited by finite lifespan, phenotypic drift during prolonged culture, and the absence of three-dimensional (3D) organization and paracrine signaling, features that may influence the establishment of persistent infections *in vivo*. In contrast, 3D organoid and microfluidic organoid-on-a-chip systems [190] are emerging as transformative tools for long-term modeling of human tissues, offering contexts that more faithfully replicate the structural, functional, and anatomical complexity of human systems (reviewed in [191,192,193]). These platforms provide opportunities to investigate how cellular organization and microenvironmental cues shape the establishment and maintenance of persistent viral infections. Notably, human brain organoids have successfully modeled persistent infection with Toscana virus and human immunodeficiency virus 1 (HIV-1) for several weeks [194,195], underscoring their potential for studying persistent stages of viral infections. However, despite these advances, current limitations include variability in differentiation protocols and incomplete incorporation of immune components, among others [191,192]. Continued development and standardization of these systems will be crucial to fully harness their potential for uncovering the mechanisms underlying alphavirus persistence.

Collectively, these advances highlight the transformative potential of **novel alternative human models** for alphavirus research. Integrating such systems with existing experimental approaches will enable more precise dissection of viral-host interactions, reveal the cellular and viral determinants of persistence versus clearance, and ultimately guide the development of therapeutic strategies targeting alphavirus infection.

## 7. Conclusions and Gaps in Knowledge

Recognition that alphaviruses can persist in host tissues long after the acute phase is reshaping our understanding of classically “acute” RNA viruses, revealing new intersections between viral infection, tissue biology, and long-term health. This emerging paradigm challenges long-held distinctions between acute and persistent viruses [66], reframing alphavirus infection as a dynamic continuum in which viral clearance, immune regulation, and tissue remodeling coexist. Despite the resolution of acute symptoms, persistence of viral RNA or antigen—documented in joints, lymphoid organs, the CNS, and, more recently, the reproductive tract—suggests that alphavirus tropism and pathophysiology might be broader than previously recognized.

As larger epidemiological datasets emerge, the broader health consequences of alphavirus infections are becoming clearer [60]. In a cohort of 143,787 individuals with confirmed CHIKV infection, the risk of death from cardiovascular causes remained elevated for up to 87 dpi, with incidence rate ratios of 2.7 and 2.4 for cerebrovascular and ischemic heart disease, respectively, at 28 dpi [60]. In addition, following the 2005–2006 CHIKV outbreak in the French territory La Réunion—which affected 38% of the island population—hypertension incidence doubled compared to mainland France, and a link with prior infection is under investigation [57]. These findings reveal potential connections between alphavirus persistence and chronic cardiovascular disease that warrant further study.

Emerging evidence that arthritogenic alphaviruses can persist and be shed in the human reproductive tract further extends this understanding, expanding the potential landscape of viral reservoirs. Persistent CHIKV RNA has been detected in the cervix [87], semen [85,196], and vaginal secretions [86] across both acute and chronic phases, suggesting reproductive tissues as possible sites of alphavirus persistence and highlighting a potential role in sexual transmission. Although human-to-human sexual transmission has not been confirmed, the isolation of VEEV and WEEV from the reproductive tissues of experimental animals following sexual activity or artificial insemination (reviewed in [197]) supports biological plausibility. These findings suggest that the reproductive tract may serve as a previously overlooked niche of alphavirus persistence and, potentially, as an alternative transmission route, raising important questions about how tissue-specific persistence contributes to both disease and viral ecology.

Mechanistically, alphavirus persistence relies on specialized cellular niches that enable partial evasion of immune clearance. Long-lived or immune-privileged cells—such as tissue-resident macrophages, fibroblasts, neurons, and FDCs—have emerged as likely reservoirs. Within these cells, alphaviruses may shift from **lytic to non-lytic infection**, reducing cytopathicity while maintaining low-level replication. The formation of DVGs and DIs, along with viral proteins such as nsP2 and capsid that modulate transcription, apoptosis, and MHC-I levels, can further promote persistence [149,151]. The host immune response, in turn, plays a dual and sometimes paradoxical role in shaping these outcomes. When immune responses become dysregulated, the balance between antiviral defense and tissue repair is disrupted. Rather than resolving after viral replication is controlled, sustained activation of innate and adaptive pathways can prolong inflammation, impair B- and T-cell effector function, and hinder tissue recovery [104]. Thus, both insufficient and excessive immune responses can drive long-term disease (Figure 3). Age, comorbidities, and genetic background can further modulate these interactions, helping explain why certain populations, such as infants, older adults, and immunocompromised individuals, are significantly affected [92,179,180,198]. Understanding these mechanisms is essential to identifying risk factors and developing targeted interventions.

Furthermore, a clearer understanding of alphavirus persistence is essential for developing targeted therapies for alphavirus-induced post-acute sequelae. Current treatments for arthritogenic-alphavirus-associated chronic arthralgia rely largely on nonspecific immunosuppression, such as via steroids, rather than mechanism-based interventions (reviewed in [199]). Determining whether persistence reflects ongoing replication or residual viral RNA or antigen will inform whether antiviral or anti-inflammatory strategies are appropriate. Identifying the factors, cellular niches, and timing that support persistence will be fundamental to guiding targeted therapeutic interventions against chronic alphavirus-induced disease.

Despite major advances, substantial knowledge gaps remain. The full spectrum of tissues and cell types capable of harboring persistent alphavirus RNA or antigen is still unclear. Moreover, how these persistent infection states are maintained for months to years within different reservoirs remains unknown. Does viral RNA continuously replicate, or is replication restricted to specific cell types? What host factors determine the duration of persistence? Are there host or environmental signals that reactivate alphavirus replication, and if so, can this lead to viremia? Furthermore, the extent to which host immune pathways—particularly interferon responses, adaptive immunity, and cytokine balance—influence viral clearance versus persistence remains elusive. Host factors such as age, immune competence, and genetic variation also appear to shape susceptibility, yet the precise mechanisms linking these variables to viral persistence require further investigation. While this review does not address persistence in other vertebrate reservoirs important in alphavirus transmission cycles (such as birds, NHPs, and small mammals), it is tempting to speculate that similar mechanisms may occur in other vertebrate hosts, potentially contributing to the maintenance of alphavirus infections within non-urban transmission cycles.

Addressing these questions will require integrative and cross-disciplinary approaches that combine single-cell and spatial technologies with animal models and advanced human *in vitro* systems, such as organoids and tissue-on-a-chip systems. Longitudinal clinical studies that monitor viral RNA or antigen persistence alongside clinical outcomes will be essential to distinguish correlation from causation in human disease. A deeper understanding of the interplay between viral replication, immune modulation, and tissue homeostasis will be key in uncovering the mechanisms that allow alphavirus persistence. Ultimately, elucidating the mechanisms by which alphaviruses persist as well as the host factors that enable or restrict this process will expand our understanding of RNA virus biology more broadly. It will also guide the development of targeted therapeutics and diagnostic tools to prevent or monitor post-acute and chronic disease outcomes. Understanding alphavirus persistence is therefore central not only to managing the long-term effects of these infections but also to improving global preparedness for emerging viral diseases.

## Figures and Tables

**Figure 2 viruses-18-00030-f002:**
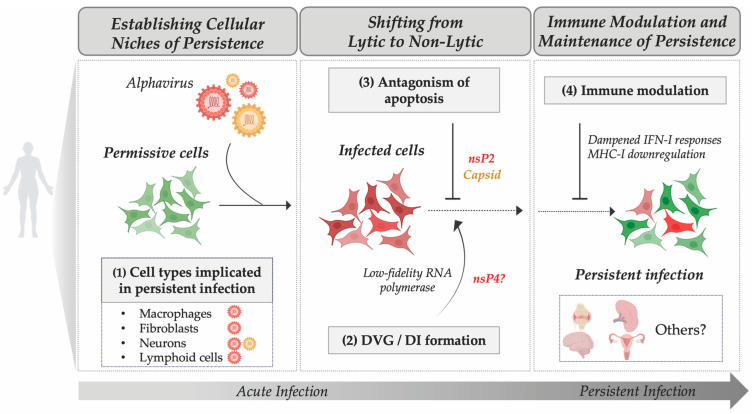
**Proposed Mechanisms Contributing to Alphavirus Persistence.** This schematic summarizes key steps hypothesized to underlie the transition from acute to persistent alphavirus infection. Persistence likely involves the following: (1) Establishment of infection in permissive or immune-privileged cell types (e.g., macrophages, fibroblasts, neurons, lymphoid cells); (2) Generation of defective viral genomes (DVGs) or defective interfering particles (DIs) that modulate replication; (3) Viral antagonism of apoptosis mediated by viral proteins such as nsP2 and capsid; and (4) Immune modulation through dampened type I interferon signaling and modulation of major histocompatibility complex class I (MHC-I). Red and green cells indicate infected and non-infected cells, respectively. Yellow particles indicate encephalitic alphaviruses. Only proposed mechanisms supporting persistent infection are shown; effective immune-mediated clearance is not depicted. Red particles indicate arthritogenic alphaviruses.

**Figure 3 viruses-18-00030-f003:**
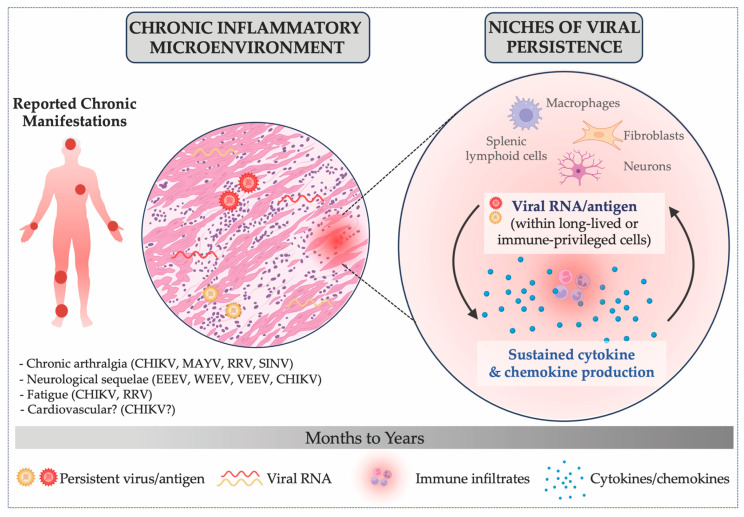
**Hypothesized Role of Persistent Alphavirus RNA and Antigen in Chronic Inflammation.** This schematic illustrates how chronic alphavirus manifestations may arise when viral RNA and antigen are not efficiently cleared from specific cellular niches. In this context, persistent viral material may sustain cytokine and chemokine production, contributing to a chronic inflammatory microenvironment characterized by ongoing immune infiltration and tissue remodeling. Some reported chronic manifestations associated with alphavirus infections include: chronic arthralgia (CHIKV [37,38,39,40], MAYV [41,42,43], RRV [44,45,46] and SINV [47,48]), fatigue (CHIKV [49,50] and RRV [51]), neurological sequelae (EEEV [35], WEEV [52,53], VEEV [54] and CHIKV [55,56]), and potential associations between alphavirus infections and long-term cardiovascular complications [57,58,59,60].

## Data Availability

No new data were created or analyzed in this study.

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
