# Peer review of "Hidden in Plain Sight: Alphavirus Persistence and Its Potential for Driving Chronic Pathogenesis"

_viruses, 2025, doi:10.3390/v18010030_

Round 1
Reviewer 1 Report
Comments and Suggestions for Authors
In this review article, the authors summarize current knowledge of persistent alphavirus infection and present careful interpretations, discuss gaps in knowledge, and provide ideas for important future directions. Overall, the manuscript is well-written, comprehensive, and should be of broad interest. Below are a few minor suggestions to improve the manuscript.
- The authors should consider eliminating the use of the terms Old- and New-World to refer to groups of alphaviruses. Although these terms have been used extensively in the alphavirus field, this is an outdated, colonial terminology that should be avoided in the opinion of this reviewer.
- Lines 83-88. PMID: 39358466 seems relevant to this section and possibly section 4.
- Line 99: Change “vertebrate hosts” to “vertebrate cells?”
- Section 2.1
- Ref 96 also reported detection of infectious virus in tissues at 44 dpi – this could be mentioned in this section.
- This section could be balanced by including studies that attempted but did not detect any evidence of viral persistence in humans, while also noting the limitations of the analysis.
- Section 2.2
- This section seems to overstate the strength of the data regarding CHIKV/RRV/MAYV persistence in humans (e.g., lines 154-155; line 165). Collectively, the data are derived from 2-3 examples.
- Section 2.3
- Lines 182-191. The following studies seem pertinent to this section: PMID: 28207896, PMID: 25474568, PMID: 27452455, PMID: 24131709
- Lines 182-191: PMID: 27736984 reported the isolation of infectious virus from joint tissues of mice at late time post-infection.
- Section 4
- The authors could expand this section to highlight studies that support the biologic plausibility of alphavirus persistence:
- Numerous studies have reported non-cytopathic alphavirus replication in various mammalian and insect cells in vitro. This is briefly mentioned, but highlighting the abundant evidence, at least in cell culture, that persistent alphavirus replication can occur may strengthen the author’s case.
- Numerous studies have reported that alphavirus replicon RNAs can persistently replicate in cell culture.
- The authors could expand this section to highlight studies that support the biologic plausibility of alphavirus persistence:
Author Response
Response to Reviewers Viruses
Reviewer #1. In this review article, the authors summarize current knowledge of persistent alphavirus infection and present careful interpretations, discuss gaps in knowledge, and provide ideas for important future directions. Overall, the manuscript is well-written, comprehensive, and should be of broad interest. Below are a few minor suggestions to improve the manuscript.
We thank the reviewer for the thoughtful and constructive evaluation of our review, which have substantially improved its quality and clarity. In this revised version, we addressed all comments from Reviewers 1 and 2 and have done additional revisions throughout the manuscript to further enhance clarity. Below, we provide a detailed point-by-point response.
1. The authors should consider eliminating the use of the terms Old- and New-World to refer to groups of alphaviruses. Although these terms have been used extensively in the alphavirus field, this is an outdated, colonial terminology that should be avoided in the opinion of this reviewer.
We agree with the reviewer and appreciate this point. We have added a clarification in the Introduction section (L30-33), and the terms “Old-World” and “New-World” were replaced with “arthritogenic” or “encephalitic” accordingly. The added text (Lines 31-39) reads:
“Although traditionally classified as “Old-World” or “New-World” alphaviruses, this geographic terminology no longer reflects their rapidly expanding global distribution. In this review, we classify alphaviruses according to their predominant clinical manifestations as arthritogenic alphaviruses – including chikungunya virus (CHIKV), Mayaro virus (MAYV), Ross River virus (RRV), Sindbis virus (SINV), Semliki Forest virus (SFV), and o’nyong-nyong virus (ONNV) – or encephalitic alphaviruses, such as Venezuelan equine encephalitis virus (VEEV), eastern equine encephalitis virus (EEEV), and western equine encephalitis virus (WEEV). Alphaviruses represent a group of human pathogens with significant epidemic potential.”
2. Lines 83-88. PMID: 39358466 seems relevant to this section and possibly section 4. We thank the reviewer for this suggestion. This study was included to L81, and incorporated in the new section 5.3 (Lines 548-550):
“Consistent with this concept, studies of respiratory paramyxovirus infection have demonstrated that long-term expression of viral antigens by surviving cells sustains innate immune activation and drives chronic pathogenesis, even in the presence of adaptive immunity.”
3. Line 99: Change “vertebrate hosts” to “vertebrate cells?” Change has been made.
4. Section 2.1:
- Ref 96 also reported detection of infectious virus in tissues at 44 dpi – this could be mentioned in this section. We have added the following text (Line 141-144):
“Interestingly, infectious particles were reported in the spleen and liver at 44 days post-infection (dpi), suggesting potential ongoing CHIKV replication. However, this interpretation is limited by the absence of mock-infected controls in the TCID50 quantifications, which cannot rule out the contribution of tissue-derived toxicity to the observed cytopathic effect.”
- This section could be balanced by including studies that attempted but did not detect any evidence of viral persistence in humans, while also noting the limitations of the analysis. To balance the section, we added the following changes (Line 121-124):
“In contrast, other studies examining synovial fluid from individuals with CHIKV-associated chronic arthralgia reported no detectable viral RNA or antigen, suggesting that, in some cases, chronic inflammation may reflect autoimmune mechanisms rather than persistent virus. Whether chronic inflammation in joint tissue after alphavirus infection is driven by persistent viral material, immune dysregulation, or both remains unresolved.”
5. Section 2.2: This section seems to overstate the strength of the data regarding CHIKV/RRV/MAYV persistence in humans (e.g., lines 154-155; line 165). Collectively, the data are derived from 2-3 examples. The following changes have been made (Line 159-161):
“In contrast to arthritogenic alphaviruses, which have several documented cases of viral RNA or antigen detection months after infection in humans, evidence of viral persistence following encephalitic alphavirus infection remains limited”
We also revised Lines 170-171 as follows:
“Unlike synovial biopsies, in which persistent viral RNA and antigen have been detected, CNS samples are rarely available for research.”
6. Section 2.3:
Lines 182-191. The following studies seem pertinent to this section: PMID: 28207896, PMID: 25474568, PMID: 27452455, PMID: 24131709. These studies have now been incorporated into the section 2.3 (Line 187-206):
“Immunocompetent mouse models of alphavirus-induced arthralgia recapitulate key features of human disease, including long-term detection of viral RNA in joint-associated tissues and the spleen. Joint fibroblasts, skeletal myofibers, macrophages, and splenic lymphoid cells have been identified as major cellular reservoirs of alphavirus RNA months after infection. However, whether this RNA represents replication-competent genomes or residual, non-replicative fragments remains under debate. The detection of positive- and negative-sense CHIKV RNA in joints up to 100 dpi, together with persistent antigen and sustained type I interferon (IFN-I) and T-cell responses, supports the possibility of ongoing active replication. Consistent with this, immunosuppression or antiviral treatment during the chronic stage of disease has revealed evidence of active viral RNA replication one month after infection for both RRV and CHIKV, with one study additionally reporting the isolation of infectious CHIKV at 90 dpi. Moreover, in recent preprint examining CHIKV infection in joint-associated tissues at single-cell resolution identified actively replicating CHIKV RNA in fibroblasts and macrophages at 28 dpi, further supporting the idea that alphaviruses can maintain long-lasting replication in specific cellular niches.
The increased severity of alphavirus-associated pathologies in young children, older adults, and individuals with comorbidities underscores the role of host immunity in shaping disease outcomes. Similar to observations in NHPs, aged mice exhibit higher levels of CHIKV RNA in joint tissues at 60 dpi compared to young adults, a difference attributed in part to elevated transforming growth factor beta (TGF-beta) levels that impair neutralizing antibody production. Consistent with the importance of adaptive immunity in controlling viral burden, mice lacking B and/or T cells develop persistent viremia, markedly elevated tissue RNA loads, pronounced inflammation, and detectable viral RNA for ≥16 weeks. Robust adaptive immune responses limit MAYV persistence, highlighting a conserved role for B- and T-cell responses across arthritogenic alphaviruses. These mechanisms are further discussed in Section 5.”
Lines 182-191: PMID: 27736984 reported the isolation of infectious virus from joint tissues of mice at late time post-infection. The following change has been made (Line 193-196):
“Consistent with this, immunosuppression or antiviral treatment during the chronic stage of disease has revealed evidence of active viral RNA replication one month after infection for both RRV and CHIKV, with one study additionally reporting the isolation of infectious CHIKV at 90 dpi.”
7. Section 4. The authors could expand this section to highlight studies that support the biologic plausibility of alphavirus persistence:
- Numerous studies have reported non-cytopathic alphavirus replication in various mammalian and insect cells in vitro. This is briefly mentioned, but highlighting the abundant evidence, at least in cell culture, that persistent alphavirus replication can occur may strengthen the author’s case.
- Numerous studies have reported that alphavirus replicon RNAs can persistently replicate in cell culture.
We thank the reviewer for this helpful comment. We have added relevant clarifications to Section 4. However, as the primary focus of this review is on mechanisms of alphavirus persistence in vertebrate hosts, and persistence in arthropod vectors involves distinct biological processes, this topic is beyond the scope of the present review. While we address the reviewer’s request, we did not include a dedicated discussion of alphavirus persistence in insects. See Lines 353–361.
“Lytic infection is driven largely by alphavirus-induced transcriptional and translational shutoff, which triggers cytopathic effects and cell death. For example, SINV nsP2 mediates host transcriptional shutdown, while VEEV relies on its capsid protein for the same function. Pioneering in vitro studies using replicon systems – later validated with full-length viruses – demonstrated that both encephalitic and arthritogenic alphaviruses can establish non-cytopathic replication in mammalian cells through adaptive mutations that reduce cytopathicity while maintaining replication. Mutations in SINV nsP2, CHIKV nsP3, and VEEV capsid protein enable persistent, non-cytopathic RNA replication in otherwise susceptible cell lines for weeks to months. Collectively, these studies provided the first mechanistic evidence that alphaviruses possess an intrinsic capacity to establish persistent infection in vertebrate cells.”
Reviewer 2 Report
Comments and Suggestions for Authors
This manuscript by Del Mar Villanueva Guzman and colleagues is a very thorough and comprehensive review of the current state of understanding of alphavirus persistence. After summarizing the current evidence of viral persistence in cells from human cases and animal models, the authors detail the features of alphaviruses that promote the ability to persist in cells and subsequently modulate the host immune response. This review is clearly written and represents an important addition to the alphavirus literature. A few suggestions are provided to further improve its depth and comprehensiveness.
- While the manuscript does a very thorough job detailing how alphaviruses establish persistence, less emphasis is placed on the second half of the title, "...Potential for Driving Chronic Pathogenesis". Even though direct evidence of persistent alphaviral RNA resulting in long-term clinical disease has not been definitively established, the manuscript would be strengthened by expanding on how the viral persistence promotes chronic inflammation, which in turn could contribute to long term clinical symptoms. Additionally, inclusion of a figure illustrating this proposed relationship would enhance the clarity and comprehensiveness of the review.
- While neurons represent an ideal cell in which alphaviruses may establish persistence in the CNS, some alphaviruses, such as CHIKV and SFV, have been shown to preferentially infect non-neuronal CNS cells, such as astrocytes and oligodendrocytes. Could the authors comment on the potential for other resident CNS cells to serve as a reservoir for viral persistence?
- Given the complex interplay between protective and pathogenic immune responses during alphavirus infection, the authors might consider adding a brief discussion on how understanding the factors driving viral persistence are important when identifying viral and host targets for therapeutics against chronic alphavirus-induced disease.
Author Response
Reviewer #2. This manuscript by Del Mar Villanueva Guzman and colleagues is a very thorough and comprehensive review of the current state of understanding of alphavirus persistence. After summarizing the current evidence of viral persistence in cells from human cases and animal models, the authors detail the features of alphaviruses that promote the ability to persist in cells and subsequently modulate the host immune response. This review is clearly written and represents an important addition to the alphavirus literature. A few suggestions are provided to further improve its depth and comprehensiveness.
We thank the reviewer for the thoughtful and constructive evaluation of our review, which has substantially improved its quality and clarity. In this revised version, we have addressed all comments from Reviewers 1 and 2 and have included additional revisions throughout the manuscript to further enhance clarity. Below, we provide a detailed point-by-point response.
1. While the manuscript does a very thorough job detailing how alphaviruses establish persistence, less emphasis is placed on the second half of the title, "...Potential for Driving Chronic Pathogenesis". Even though direct evidence of persistent alphaviral RNA resulting in long-term clinical disease has not been definitively established, the manuscript would be strengthened by expanding on how the viral persistence promotes chronic inflammation, which in turn could contribute to long term clinical symptoms. Additionally, inclusion of a figure illustrating this proposed relationship would enhance the clarity and comprehensiveness of the review.
We agree with the reviewer and, to expand on potential scenarios in which viral persistence may promote chronic inflammation, we included a new Section 5.3, and a new Figure 3 (Lines 538-553):
“5.3 Persistence as a Potential Driver of Chronic Pathogenesis. Adaptive immunity is required, yet often insufficient, for complete clearance of alphaviruses. Even in the presence of robust humoral and cellular responses, viral RNA and antigen can persist in select tissues, including joint, lymphoid, muscle, reproductive, and CNS compartments. Such persistence reflects a finely balanced interplay between viral immune evasion, tissue-specific immune privilege, and dysregulated cytokine activity. In this context, immune activation and viral immune escape coexist, illustrating how pathways that normally restrict viral replication may, when dysregulated, sustain chronic inflammation (Figure 3). Although direct causal evidence linking persistent alphavirus RNA to long-term pathologies remains lacking, a growing body of experimental and clinical observations suggests that viral persistence may maintain inflammatory programs that contribute to chronic disease. Consistent with this concept, studies of respiratory paramyxovirus infection have demonstrated that long-term expression of viral antigens by surviving cells sustains innate immune activation and drives chronic pathogenesis, even in the presence of adaptive immunity. Supporting a similar mechanism in alphavirus infection, a recent preprint examining CHIKV-infected mouse joints showed that treatment with a small-molecule inhibitor of viral replication reduced both CHIKV RNA levels and inflammatory gene signatures. These findings provide the first experimental indication that persistent alphavirus RNA arising from ongoing low-level replication may contribute to chronic inflammation. Therapeutically, this duality highlights the need for strategies that promote effective adaptive viral clearance while minimizing the risk of exacerbating inflammatory damage.”
Figure 3. Hypothesized Role of Persistent Alphavirus RNA and Antigen in Chronic Inflammation. This schematic illustrates how chronic alphavirus manifestations may arise when viral RNA and antigen are not efficiently cleared from specific cellular niches. In this context, persistent viral material may sustain cytokine and chemokine production, contributing to a chronic inflammatory microenvironment characterized by ongoing immune infiltration and tissue remodeling. Some reported chronic manifestations associated with alphavirus infections include: chronic arthralgia (CHIKV, MAYV, RRV and SINV), fatigue (CHIKV and RRV), and neurological sequelae (EEEV, WEEV, VEEV and CHIKV), and potential associations between alphavirus infections and long-term cardiovascular complications.
2. While neurons represent an ideal cell in which alphaviruses may establish persistence in the CNS, some alphaviruses, such as CHIKV and SFV, have been shown to preferentially infect non-neuronal CNS cells, such as astrocytes and oligodendrocytes. Could the authors comment on the potential for other resident CNS cells to serve as a reservoir for viral persistence?
We thank the reviewer for this suggestion. The following paragraph was added to lines 314-317.
“Non-neuronal CNS cells such as oligodendrocytes and astrocytes are also long lived and can be infected by arthritogenic alphaviruses both in vitro and in vivo. The detection of persistent SFV antigen within astrocytes in infected immunodeficient mice further highlights the potential for these glial cell populations to contribute to alphavirus persistence within the CNS.”
3. Given the complex interplay between protective and pathogenic immune responses during alphavirus infection, the authors might consider adding a brief discussion on how understanding the factors driving viral persistence are important when identifying viral and host targets for therapeutics against chronic alphavirus-induced disease. We thank the reviewer for this helpful comment. We have added a paragraph in the discussion section addressing this point. See lines 649-654:
“Furthermore, a clearer understanding of alphavirus persistence is essential for developing targeted therapies for alphavirus-induced post-acute sequelae. Current treatments for arthritogenic-alphavirus-associated chronic arthralgia rely largely on nonspecific immunosuppression, such as via steroids, rather than mechanism-based interventions. Determining whether persistence reflects ongoing replication or residual viral RNA or antigen is critical for choosing antiviral or anti-inflammatory strategies. Identifying the factors, cellular niches, and timing that support persistence will be fundamental to guiding targeted therapeutic interventions against chronic alphavirus-induced disease.”